

# Sub-Doppler laser cooling of $^{39}$K via the 4S→5P transition

**Govind Unnikrishnan$^\star$, Michael Gröbner and Hanns-Christoph Nägerl**

Institut für Experimentalphysik und Zentrum für Quantenphysik,
Universität Innsbruck, 6020 Innsbruck, Austria

$\star$ govind.unnikrishnan@uibk.ac.at

## Abstract

We demonstrate sub-Doppler laser cooling of $^{39}$K using degenerate Raman sideband cooling via the $4S_{1/2} \to 5P_{1/2}$ transition at 404.8 nm. By using an optical lattice in combination with a magnetic field and optical pumping beams, we obtain a spin-polarized sample of up to $5.6 \times 10^7$ atoms cooled down to a sub-Doppler temperature of 4 μK, reaching a peak density of $3.9 \times 10^9$ atoms/cm$^3$, a phase-space density greater than $10^{-5}$, and an average vibrational level of $\langle v \rangle = 0.6$ in the lattice. This work opens up the possibility of implementing a single-site imaging scheme in a far-detuned optical lattice utilizing shorter wavelength transitions in alkali atoms, thus allowing improved spatial resolution.

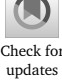

# 1  Introduction

Sub-Doppler laser cooling is an important intermediate step between Doppler cooling [1–3] and evaporative cooling in most quantum gas experiments. The ability to reach ultralow temperatures and quantum degeneracy by these techniques has heralded a whole series of experiments, such as precision measurements [4, 5], quantum simulations [6, 7], studies of out-of-equilibrium many-body dynamics [8] and impurity dynamics measurements in low dimensions [9–12]. More recently, sub-Doppler cooling schemes have also been utilized for performing single-site high-resolution imaging in deep optical lattices, wherein the photons scattered during the cooling process are collected to reveal in-situ images with single-site resolution [13–20]. Such measurements reveal a wealth of information about the system that is not accessible via time-of-flight techniques, e.g. spatial and spin correlations.

Raman sideband cooling, initially implemented for ions [21] and later for atoms in optical lattices [16, 19, 22–28] and tweezers [29–31], is a powerful sub-Doppler cooling technique by which trapped atoms can be imaged and cooled to high phase-space densities. An additional advantage of Raman sideband cooling over molasses cooling, which was initially employed for high-resolution imaging [13, 14], is that it maintains a large fraction of atoms in the motional ground state of a lattice site [16, 19]. It has been successfully used to cool Cs [22, 23, 23–25], Rb [26, 27, 32], K [16, 33] and Li [19] to sub-Doppler temperatures in optical lattices. Both Li and K are attractive choices for ultracold quantum gas experiments since along with bosonic isotopes, there also exist stable fermionic isotopes for these species. While sub-Doppler cooling schemes such as traditional molasses cooling are not effective for these species due to the closely spaced excited hyperfine levels [34, 35], Raman sideband cooling has been proven to work successfully for K and Li. However, for these elements, the $n\text{S}_{1/2} \to n\text{P}_{1/2}$ transitions are utilized since the $n\text{P}_{1/2}$ hyperfine levels have larger hyperfine splittings compared to the corresponding linewidths, thus reducing off-resonant excitations detrimental to the cooling process.

Transitions with narrow linewidths have previously been used to perform Doppler cooling, since the Doppler temperature $T_{\text{D}} = \hbar\Gamma/2k_{\text{B}}$ scales with the linewidth $\Gamma$ of the cooling transition. Such narrow-line cooling has been demonstrated on dipole-forbidden transitions in Sr and Ca [36–38]. The $n\text{S} \to (n+1)\text{P}$ transition in Li and K, which has a linewidth smaller than the $n\text{S} \to n\text{P}$ transition, has been used to cool and trap atoms in a magneto-optical trap (MOT) to higher densities and lower temperatures than typical MOTs working on the $n\text{S} \to n\text{P}$ transition [35, 39]. Another advantage of using transitions to higher excited states is provided by the wavelengths associated with such transitions. The use of a short wavelength can provide a strong improvement given the diffraction limit, which can prove to be particularly useful for single-site high-resolution imaging schemes. Additionally, the large separation of higher excited state transition wavelengths from those used for typical cooling schemes can improve background-light rejection, leading to better signal-to-noise ratios. However, an important distinction of such a transition is that it involves spontaneous decays into multiple levels from the excited state. The concomitant depolarization of atomic states associated with such a decay mechanism can hamper the efficiency of laser cooling mechanisms. Cooling techniques on such transitions have thus far produced temperatures well above the corresponding Doppler temperatures.

In this work, we demonstrate sub-Doppler laser cooling of $^{39}\text{K}$ using degenerate Raman sideband cooling (dRSC) via the $4\text{S}_{1/2} \to 5\text{P}_{1/2}$ transition at 404.8 nm. We polarize up to $5.6 \times 10^7$ atoms into the $|F = 2, m_F = -2\rangle$ state and reach temperatures below 4 μK, which corresponds to $0.15\ T_{\text{D}}$ and $0.03\ T_{\text{D}}$ for the $5\text{P}_{1/2}$ and $4\text{P}_{1/2}$ states, respectively. We reach a peak density of $3.9 \times 10^9$ atoms/cm$^3$ at which our phase-space densities are larger than $10^{-5}$. We also determine a mean vibrational quantum number of $\langle v \rangle = 0.6$ in the lattice. Finally, to

demonstrate the feasibility of implementing high-resolution imaging schemes on excited state transitions, we directly collect the 404.8-nm fluorescence and estimate a scattering rate of 2.2 kHz per atom.

## 2  Background

### 2.1  The 4S→5P transition

The energy level diagram with all relevant states between the $4S_{1/2}$ ground state and the $5P_{1/2}$ excited state for $^{39}K$ is shown in Fig. 1(a). Populating the $(n+1)^{th}$ excited state has several consequences. As opposed to the more prevalently utilized cyclic transitions between the $n$S and $n$P levels, the $n$S→$(n+1)$P transitions involve a radiative cascade to various intermediate states, as shown in Fig. 1(a). Laser cooling in alkali atoms typically involves cycling between one of the two hyperfine ground states and an excited state, usually with a repumping light field on the other hyperfine state. Raman cooling techniques on the $n$S→$n$P transitions have been adapted to prevent populating the second hyperfine level except via off-resonant excitations by working on the $F_L \rightarrow F' = F_L - 1$ closed transition ($F_L$ being the lower hyperfine ground state), since a direct decay $F' = F_L - 1 \rightarrow F_U$ ($F_U$ being the upper hyperfine state) is forbidden [23, 32]. However, the cascading spontaneous decay channels involved with $n$S→$(n+1)$P transitions always allow both hyperfine ground states to be populated. Furthermore, even within a given hyperfine state, various Zeeman sublevels can get populated. Although such a depolarization of Zeeman states may occur even when working with the $n$S→$n$P transitions [33], the radiative cascade makes this a much stronger process. For cooling schemes that depend on preferentially shelving the atomic population into a particular $m_F$ state, this process becomes particularly relevant.

We quantify this effect when driving the $4S_{1/2} \rightarrow 5P_{1/2}$ transition and compare it to the $4S_{1/2} \rightarrow 4P_{1/2}$ transition used in Ref. [33]. First, the possible decay routes by which an atom in the $5P_{1/2}|F = 1, m_F = -1\rangle$ state can reach the different Zeeman levels in the ground-state manifold are identified. We obtain the probabilities for each transition by normalizing each $J' \rightarrow J$ transition rate by the sum of the transition rates along all possible decay channels from a given state and scaling it with the Clebsch-Gordon coefficients for the corresponding $|F, m_F\rangle \rightarrow |F', m_F'\rangle$ transitions. Summing the probabilities along all the possible decay routes from the $5P_{1/2}$ level to the different Zeeman levels in the ground state, we obtain the results shown in Fig. 1(b). We also calculate these probabilities when an atom excited to the $|F = 1, m_F = -1\rangle$ state in the $4P_{1/2}$ level undergoes spontaneous decay. For the $|F = 2, m_F = -2\rangle$ ground state, we obtain a probability of 34% and 50% for the blue ($4S_{1/2} \rightarrow 5P_{1/2}$) and the red ($4S_{1/2} \rightarrow 4P_{1/2}$) transitions, respectively. The fact that we get a significant population into the $|F = 2, m_F = -2\rangle$ state despite the radiative cascade is an indication that a dRSC scheme can work on such a transition, as we verify in this work.

Other factors that distinguish the $(n+1)^{th}$ excited state from the $n^{th}$ excited state are their small linewidth, high saturation intensity, small absorption cross section and the possibility of photoionization from excited states. In view of the fact that laser cooling schemes on the 4S→4P lines of K [34, 41–44] are often limited by the small hyperfine splittings of the excited states, the hyperfine splitting of the $5P_{1/2}$ level (18.6 MHz) [45] is 16.5 times larger than its linewidth (1.13 MHz) [40], providing better state selectivity for sub-Doppler cooling schemes. On the other hand, the saturation intensity for the $5P_{1/2}$ level is more than 10 times larger than the $4P_{1/2}$ level (56 mW/cm$^2$ compared to 5.1 mW/cm$^2$ for unpolarized light) and one would require higher intensities to achieve comparable scattering rates. One may also expect lower density-dependent heating due to rescattering of light with an absorption cross section
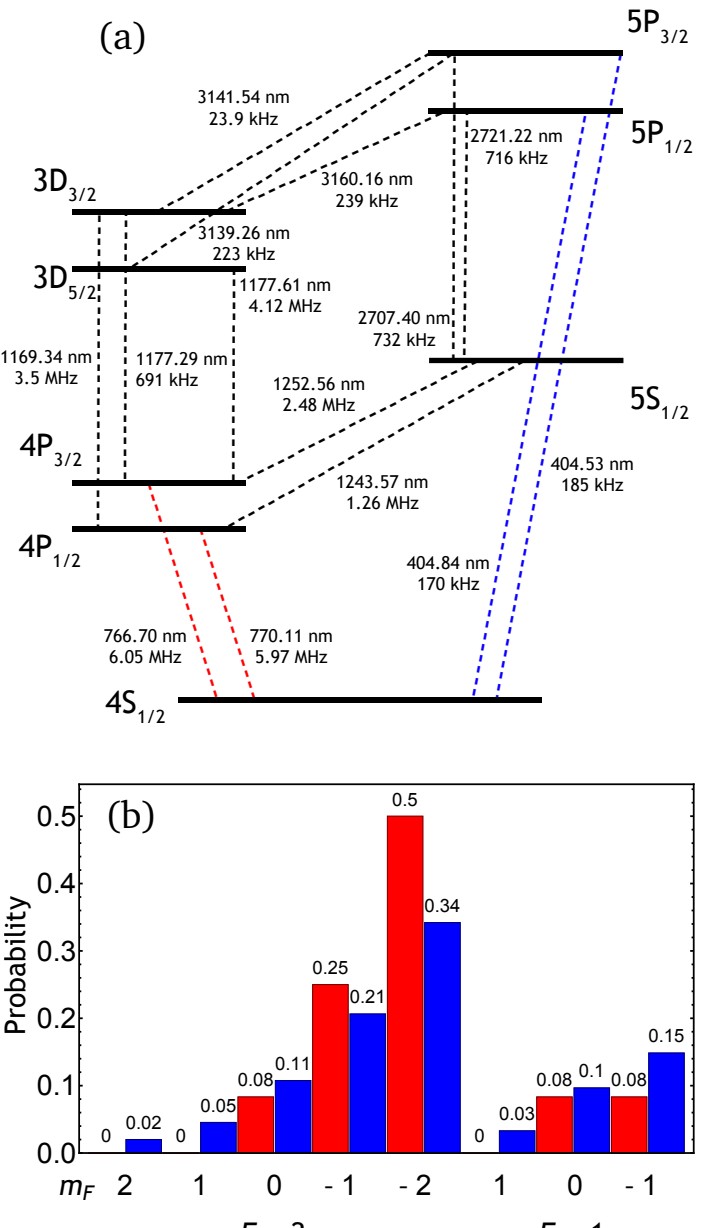

Figure 1: (a) Energy level diagram with all allowed transitions between the $4S_{1/2}$ and $5P_{3/2}$ levels. Corresponding wavelengths and individual linewidths $\Gamma/2\pi$ are also shown [40]. For states with multiple decay channels, the effective linewidth is given by the sum of the linewidths of each possible decay from that state. Dashed red lines indicate typical transitions used for laser cooling in $^{39}$K and dashed blue lines indicate the two possible transitions close to 404 nm. (b) Depolarization into different Zeeman states in the hyperfine ground states $F = 2$ and $F = 1$ due to radiative cascade. The red bars indicate the probabilities for decay from the $4P_{1/2}|F = 1, m_F = -1\rangle$ (relevant for the dRSC scheme in Ref. [33]) and the blue bars from the $5P_{1/2}|F = 1, m_F = -1\rangle$ state.

20 times smaller than the 4S→4P line.

Finally, we discuss the possibility of photoionization. The ionization threshold from the levels $4S_{1/2}$, $4P_{1/2}$, $4P_{3/2}$, $5S_{1/2}$, $3D_{3/2}$, and $5P_{1/2}$ is 286 nm, 454 nm, 455 nm, 715 nm, 742 nm and 972 nm away, respectively [40]. The photoionization rate from the state $S$ scales as $\gamma_S = \rho_S \sigma_S \frac{I\lambda}{hc}$, where $\rho_S$ is the fraction of atoms in the state $S$, $\sigma_S$ is the ionization cross section, $I$ the intensity of light and $\lambda$ the wavelength [46]. The polarizer can cause ionization from all levels except the ground state, whereas the repumper and the lattice light can only ionize the atoms in the 5P excited state (404.8 nm being the wavelength of the polarizer, whereas the repumper and lattice light fields are close to 770 nm and 767 nm, respectively). At the intensities and detunings used in our experiment, we calculate a total ionization rate that is less than 100 mHz, which should not play an important role during our cooling time of about 5 ms. We also note that for typical far-detuned lattices at 1064 nm relevant for high-resolution imaging, no photoionization is possible except via a multi-photon process.

## 2.2 Degenerate Raman sideband cooling

Our dRSC scheme builds upon previous work on Cs and K [22–25, 33]. In all previous cooling schemes, optical pumping was carried out on the $n$S→ $n$P transition. The distinguishing feature of our method is the optical pumping on the $n$S→ $(n + 1)$P transition. The cooling scheme works on the ground-state $|F = 2\rangle$ manifold and is illustrated in Fig. 2. Atoms are trapped in the micropotentials created by an optical lattice where their center-of-mass motion is quantized. A magnetic field is applied to induce a Zeeman splitting between the different $m_F$ levels and to bring the $|F = 2, m_F, \nu\rangle$ state into degeneracy with the $|F = 2, m_F + 1, \nu - 1\rangle$ state, where $\nu$ denotes the vibrational quantum number. The frequency and polarizations of the lattice light beams are chosen such that there are significant Raman couplings due to the lattice light between the states that are brought into degeneracy by the magnetic field [22, 33]. An atom starting from a higher vibrational state $\nu$ in a state $m_F$ undergoes Raman transitions into a vibrational level $\nu - 1$ in the $m_F + 1$ level. Atoms starting from the $|F = 2, m_F = -2, \nu\rangle$ state thus reach the $|F = 2, m_F = 0, \nu - 2\rangle$ state, from which they are optically pumped by a $\sigma^-$-polarized beam tuned to the $4S_{1/2}|F = 2\rangle \rightarrow 5P_{1/2}|F = 1\rangle$ transition (so called "polarizer"). In the Lamb-Dicke regime, where the recoil energy $E_R$ due to optical pumping is much smaller than the vibrational energy $\hbar\omega$ in the lattice ($\omega$ being the harmonic oscillation frequency at a lattice site), the vibrational quantum number is preserved during spontaneous emissions. Hence, an atom excited from the $|F = 2, m_F = 0, \nu - 2\rangle$ state undergoes a decay into the $|F = 2, m_F = -2, \nu - 2\rangle$ state with the highest probability, hence losing two vibrational quanta of energy. A $\pi$-polarized component of the polarizer beam, obtained by slightly tilting the quantization axis, pumps the atoms from the $|F = 2, m_F = -1, \nu = 0\rangle$ state into the $|F = 2, m_F = -2, \nu = 0\rangle$ state. After multiple such cooling cycles, we shelve a large portion of the atomic population into the $|F = 2, m_F = -2, \nu = 0\rangle$ state, which is dark to the optical pumping beams. Note that a second $\sigma^-$-polarized beam (so called "repumper"), which is tuned to the $4S_{1/2}|F = 1\rangle \rightarrow 4P_{1/2}|F = 2\rangle$ transition (see Fig. 3), depletes population in the $4S_{1/2}|F = 1\rangle$ manifold.

# 3 Experiment

## 3.1 Setup

Our experimental setup and the dRSC scheme, including the generation and frequency shifting of the lattice (80 mW, 663 mW/cm$^2$) and repumper (0.6 mW, 0.7 mW/cm$^2$) light, are as described in Refs. [33, 47]. At 663 mW/cm$^2$ lattice intensity and our optimal lattice detuning

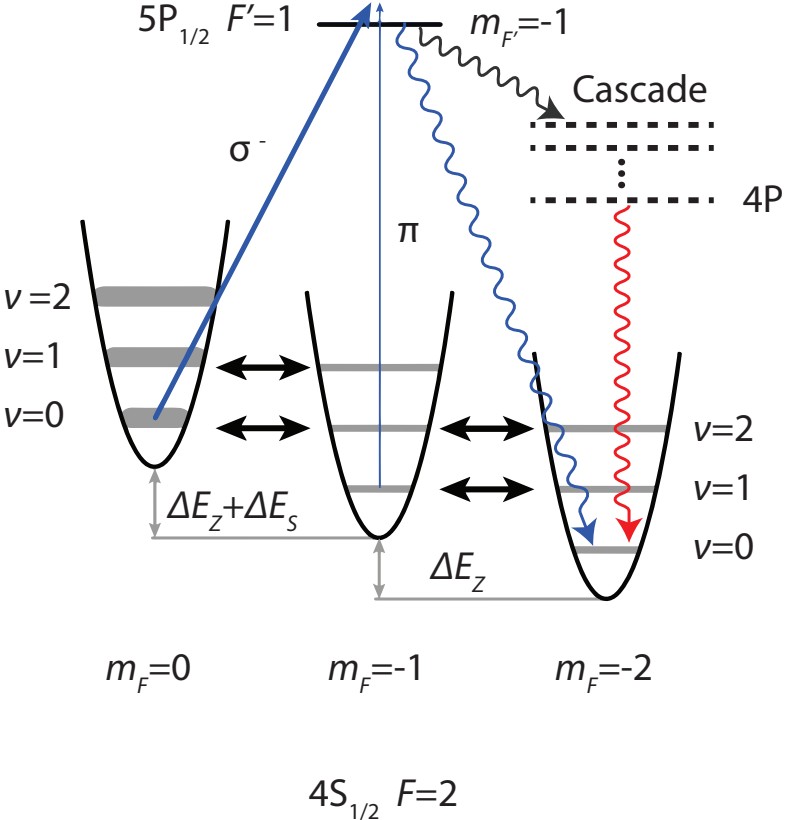

Figure 2: The degenerate Raman sideband cooling scheme via the $4S_{1/2} \rightarrow 5P_{1/2}$ transition. $\Delta E_Z$ is the Zeeman splitting, which brings the $|F = 2, m_F, v\rangle$ state into degeneracy with the $|F = 2, m_F + 1, v - 1\rangle$ state. The double-sided arrows indicate Raman transitions between these states, which are provided by the optical lattice. $\Delta E_S$ is the light shift of the polarizer beam with $\sigma^-$ polarization. The broadening of vibrational levels due to the polarizer beam is indicated by the thick grey lines. Optical pumping from the $m_F = -1$ and $m_F = 0$ states is indicated by blue arrows. The blue wiggled line indicates direct spontaneous decay from the $5P_{1/2}$ level by emitting a blue photon, the black wiggled line represents the decay into intermediate states, and the red wiggled line indicates the decay from the 4P manifold to the ground state. Details of the repumping process are given in Ref. [33].



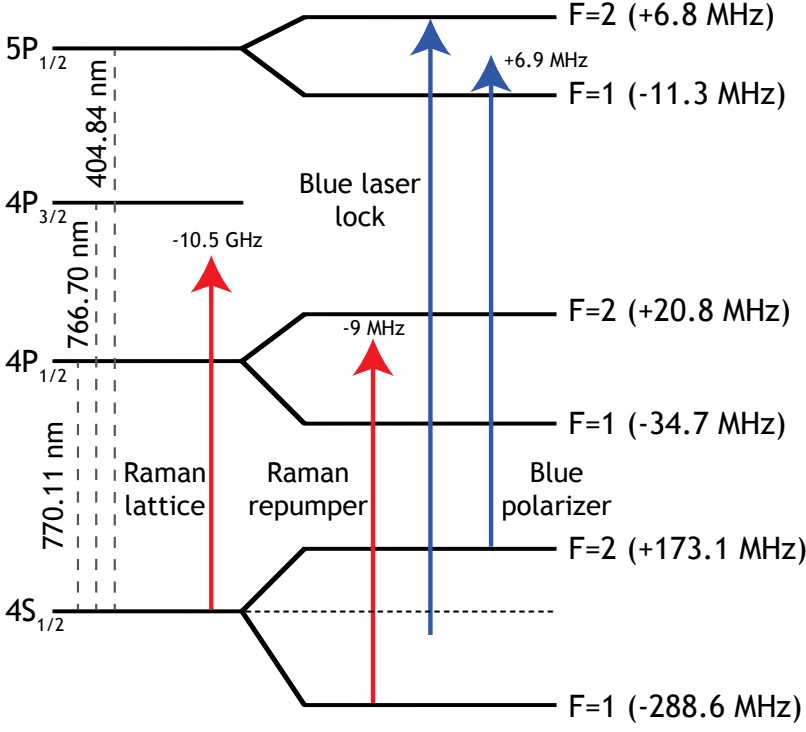

Figure 3: Energy level diagram with the hyperfine structure and the relevant transitions. For optimum performance, we use the Raman lattice laser 10.5 GHz red detuned from the $4P_{3/2}$ state, the polarizer 6.9 MHz blue detuned from the $5P_{1/2}|F=1\rangle$ state, and the repumper 9 MHz red detuned from the $4P_{1/2}|F=2\rangle$ state. The ground-state crossover to the $5P_{1/2}|F=2\rangle$ transition to which the 404-nm laser is locked is also shown.

of -10.5 GHz, the lattice depths are $(U_x, U_y, U_z) \approx (61.3, 29.7, 29.7)$ μK. We use the same beam configuration as in Ref. [33] except that the polarizer and repumper beams are sent vertically upwards along the direction of the quantization axis. A home-built external-cavity diode laser in Littrow configuration provides the light at 404.8 nm required for the polarizer. We use an uncoated Nichia laser diode with a peak wavelength at 405 nm stabilized to a temperature of 37.4°C operated at a total output power of 16 mW. The laser diode is operated within a two-chamber housing that is also stabilized in temperature. The diode is shielded from acoustic noise by keeping the entire setup within a padded wooden box. Our mode-hop free regime spans more than 500 MHz. We observe a drift of this regime over several days, which we compensate by slightly changing the set temperature of the diode. The laser is locked from the ground-state crossover to the $5P_{1/2}|F=2\rangle$ level via modulation-transfer spectroscopy. For spectroscopy, we use a total power of 2 mW and keep the K vapor cell at around 70°C. We shift the frequency to the $4S_{1/2}|F=2\rangle$ to $5P_{1/2}|F=1\rangle$ transition via a double-pass acousto-optic modulator (AOM) driven close to 124 MHz. We show the relevant transitions and detunings in Fig. 3. Before reaching the atoms, the polarizer beam is overlapped with the repumper and the vertical lattice beam via a dichroic mirror. The polarizer beam is slightly focused so that it has a beam size of 3.2 mm at the position of the atomic cloud, corresponding to a maximum intensity of 2.9 mW/cm$^2$.

## 3.2 Measurements

Our measurements begin by loading $2.5 \times 10^8$ $^{39}$K atoms from a 2D$^+$ MOT into a 3D MOT in 5 s. We then proceed to the compressed MOT stage and gray molasses cooling (GMC) as described in Ref. [33]. After GMC, we obtain an unpolarized sample of $2.5 \times 10^8$ atoms in the $|F = 1\rangle$ hyperfine ground state. Although we can reach a temperature of 6 μK at this stage, we purposefully stop cooling when the cloud is 12 μK to demonstrate the cooling efficiency of the upcoming dRSC stage. We turn on the magnetic offset field required to create the relative shifts between the different $m_F$ levels and the lattice beams 1.5 ms prior to the end of the GMC stage. Immediately afterwards, the repumper and the polarizer beams are turned on at detunings of -9 MHz and 6.9 MHz, respectively. In order to ensure efficient loading of the optical lattice by cooling initially unbound atoms into the lattice [23, 32], the magnetic offset field is kept at a value slightly larger than the one used later during the initial 0.5 ms of the cooling period. The offset field is then ramped down to an experimentally optimized value in 0.5 ms and then slowly ramped up in 5 ms so that multiple $|F, m_F, \nu\rangle$-$|F, m_F + 1, \nu - 1\rangle$ energy levels pass through degeneracy during the ramp (the actual magnetic fields used are very close to those estimated in Ref. [33]). Finally, the atoms are adiabatically released from the lattice by ramping down the lattice power in 0.35 ms. The repumper and polarizer are also ramped down during this period.

In order to probe the temperature of the $|F = 2, m_F = -2\rangle$ component, we do Stern-Gerlach separation by applying a magnetic-field gradient of 8 G/cm and an offset field of 20 G for 30 ms. Due to reasons explained in Ref. [33], we obtain an upper bound on the temperature of the $|F = 2, m_F = -2\rangle$ spin component of the cloud by measuring the horizontal time-of-flight (TOF) expansion via absorption imaging. We measure a threefold reduction in temperature to 0.15 $T_D$(4 μK) and observe up to $5.6 \times 10^7$ atoms in the $|F = 2, m_F = -2\rangle$ spin state corresponding to a phase-space density of $1.1 \times 10^{-5}$. We estimate systematic errors less than 25% in our temperature and atom number measurements. The statistical error is negligible given the systematic error.

In Fig. 4(a), we show how the cooling proceeds over time. We vary the duration of the final ramp up of the magnetic field, $t_C$, and measure the resulting temperature $T$ by recording the cloud radius $\sigma_X$ (standard deviation of a Gaussian fit to the integrated optical density) from different TOF images. The temperature drops exponentially with $t_C$ and saturates at around 4 μK after 5 ms of cooling. A temperature of 4 μK corresponds to a cloud radius of about 1.4 mm after TOF expansion of 37 ms. The temperature achieved by dRSC on the $4S_{1/2} \rightarrow 4P_{1/2}$ transition (red dRSC) is 1.8 μK [33]. We attribute two main reasons for the higher temperature. Firstly, due to the depolarization into the various $m_F$ states described earlier, an atom would have to undergo many more scattering events on average before reaching the dark state $|F = 2, m_F = -2, \nu = 0\rangle$ when compared to the red dRSC scheme. Each scattering event into an $m_F$ state without loss of vibrational quanta leads to heating by at least two photon recoils $E_R$, where $E_R = \hbar^2 k^2 / 2m$ ($k = 2\pi/\lambda_T$ is the wave vector, $\lambda_T$ being the wavelength of the transition; $m$ is the mass of the atom). In addition, the recoil energy associated with a 404-nm photon is 3.6 times larger compared to a 770 nm photon. Secondly, the increased probability to decay into the $F = 1$ hyperfine level and the multiple levels involved in the radiative cascade leads to a larger steady state scattering rate as compared to the red dRSC scheme. During the time spent in the lower hyperfine state, atoms get heated as vibrational levels are more tuned towards the first blue sideband in that manifold [32]. Furthermore, photon scattering during the repumping process also leads to heating.

Fig. 4(a) also shows how the mean vibrational quantum number $\langle \nu \rangle$ varies as we cool down the atoms. Assuming a thermal distribution of atoms in the harmonic potentials, the mean vibrational quantum number is obtained as $\langle \nu \rangle = \frac{1}{\exp(\hbar\omega/k_B T) - 1}$ [48], $\omega$ being the angular trap frequency at a lattice site and $T$ the temperature. We obtain the temperature $T$ via TOF

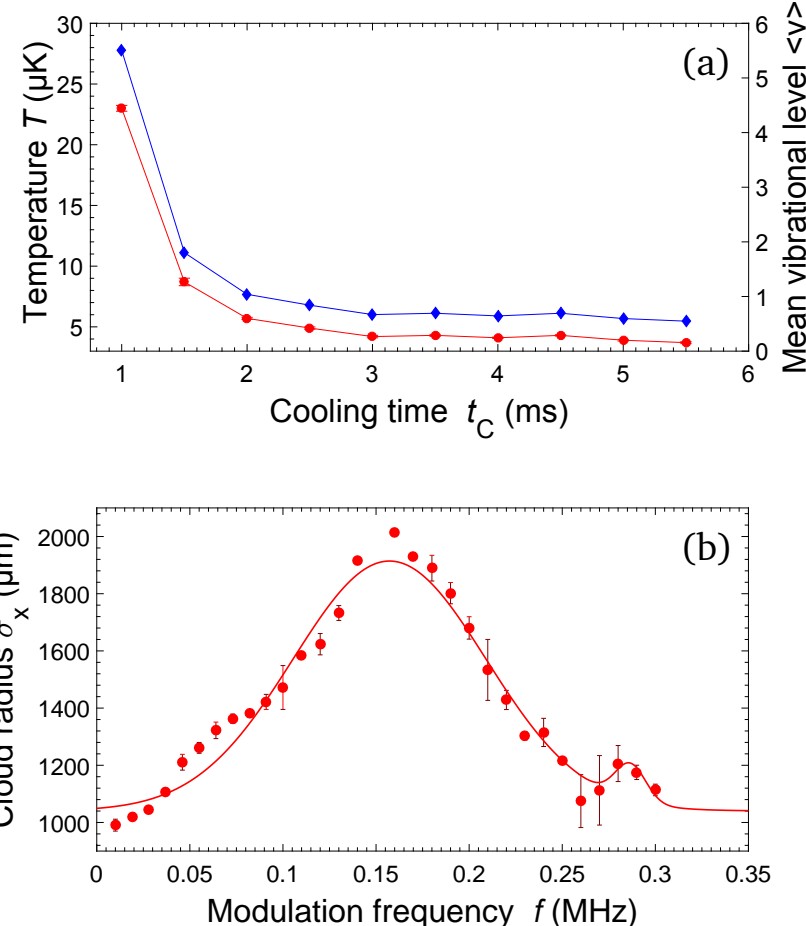

Figure 4: (a) Dependence of the temperature $T$ (red circles) and mean vibrational level $\langle v \rangle$ (blue diamonds) on the cooling duration $t_C$. After 5 ms of cooling, we reach a temperature of 4 μK. A polarizer intensity of 2.9 mW/cm$^2$ corresponding to 115 μW of power is used. (b) Parametric excitation spectrum obtained by modulating the lattice intensity by 26% for 5 ms at a frequency $f$. We observe resonant heating, corresponding to an increased cloud radius, when the modulation frequency matches twice the trap frequency. The data is fitted to a double Gaussian function, which yields centre frequencies of 160 and 277.5 kHz. We use a total lattice power $P_L = 80$ mW. Detunings are indicated in Fig. 3. Error bars indicate standard deviations in both (a) and (b).

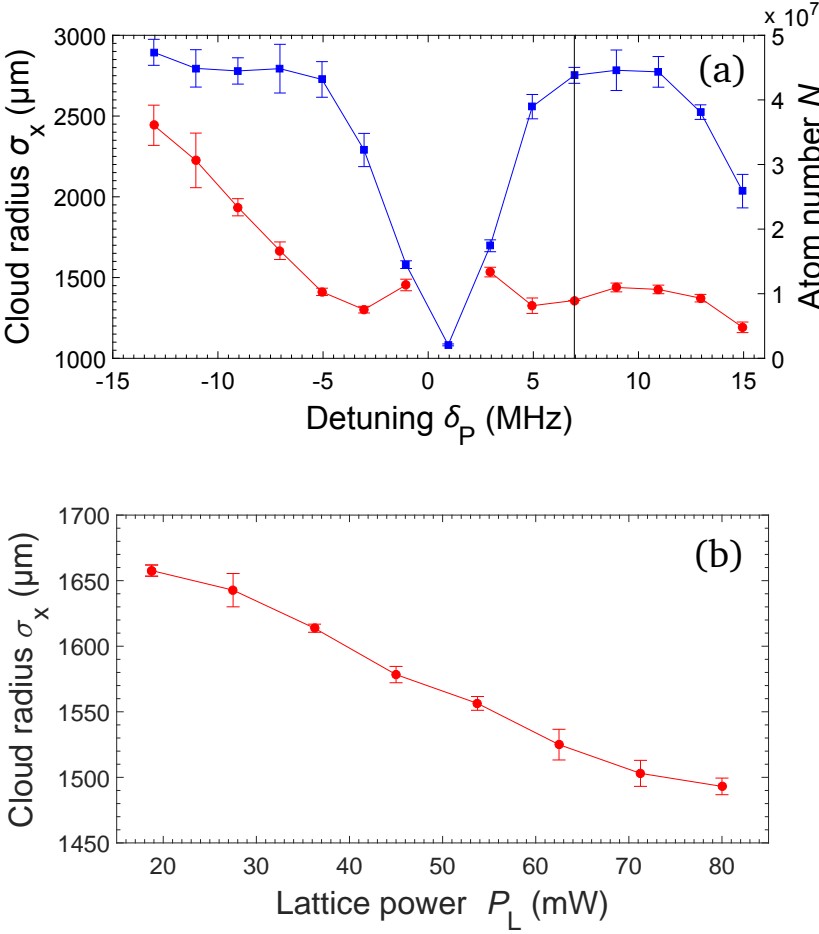

Figure 5: (a) Cloud radius $\sigma_X$ (red circles) and atom number (blue squares) post cooling as a function of the polarizer detuning $\delta_P$. We use a total lattice power $P_L = 80$ mW, a polarizer intensity of 2.9 mW/cm$^2$ corresponding to 115 μW of power and a repumper power of 0.6 mW at -9 MHz detuning. (b) Cloud radius as a function of the total power in the three lattice beams. Polarizer and repumper parameters are the same as in (a) except that the polarizer detuning is kept at 6.9 MHz (vertical line). Error bars indicate standard deviations in both (a) and (b).

expansion measurements as before. To measure $\omega$, we record a parametric excitation spectrum by modulating the lattice intensity by 26% for 5 ms and recording the observed heating [24, 32, 49]. For 80 mW of lattice power (663 mW/cm$^2$ intensity) at -10.5 GHz detuning, we observe a significant increase in the cloud radius at a modulation frequency of 160 kHz and a smaller increase at 277.5 kHz as shown in Fig. 4(b). Since maximum resonant heating is observed at twice the trap frequency [49], we infer trap frequencies of $\omega_x/2\pi = 138.8$ kHz and $\omega_{y,z}/2\pi = 80$ kHz, $x$ being the direction of the standing wave and $y, z$ being the directions along which the other two lattice beams propagate, as described in Ref. [33]. Close to 70 kHz, we observe a slight increase in cloud radius, which we attribute to heating at a harmonic of the trap frequency. As the Raman couplings of $(\Omega_x, \Omega_y, \Omega_z)/2\pi \approx (0, 45, 45)$ kHz used in our experiment are comparable to the trap frequencies, the three vibrational degrees of freedom are strongly mixed by the Raman transitions [32], leading to strong broadening of the peaks in our excitation spectrum. Assuming an effective trap frequency of 80 kHz, we calculate a mean vibrational quantum number of $\langle v \rangle = 0.6$ at a temperature of 4 μK.

The dependence of the cooling process on the polarizer detuning $\delta_\text{P}$ is shown in Fig. 5(a). We perform the entire experimental sequence for different values of $\delta_\text{P}$ and record the final atom number and the cloud radius. The number of atoms that can be cooled and the final achievable temperature has a strong dependence on $\delta_\text{P}$. Close to resonance, a strong power broadening and light shift takes the system out of the configuration where efficient cooling is possible. We observe optimal cooling performance at 6.9 MHz blue detuning. For red detunings with reasonable atom numbers, we observe a similar atom number as for commensurate blue detunings but significantly higher temperatures, as shown in Fig. 5(a). We explain this as follows. The $\sigma^-$-polarized polarizer beam shifts the $m_F = 0$ level up and down for blue and red detuning of the polarizer, respectively. For red detunings, this increases the possibility of Raman transitions with $\Delta \nu = 0$ between the $m_F = 0$ and $m_F = -1$ levels, which in turn can lead to heating. On the other hand, the upward shift due to blue detuning enhances the initial cooling into the lattice [32]. For larger blue detunings, we observe a decrease in atom number, which we attribute to the increased probability of excitation to the $F = 2$ state of the $5\text{P}_{1/2}$ level.

Finally, we record the radius of the cloud after cooling as a function of the lattice power $P_\text{L}$. Our measurements are shown in Fig. 5(b). We observe a consistent decrease in the final cloud radius (corresponding to a temperature decrease) with increasing lattice power. A higher lattice power allows one to stay deeper in the Lamb-Dicke regime during the cooling process, thus enabling more efficient cooling. The Lamb-Dicke parameters, defined as $\eta_i = \sqrt{\frac{E_\text{R}}{\hbar \omega_i}}$ with $i = x, y, z$ and $E_\text{R}$ the recoil energy of a 404.8-nm photon, corresponding to a lattice intensity of 663 mW/cm$^2$ and a -10.5-GHz detuning, are $(\eta_x, \eta_y, \eta_z) = (0.47, 0.63, 0.63)$ in the three trapping directions. Another important distinction with respect to the red dRSC scheme is the 5.4 times larger lifetime of the $5\text{P}_{1/2}$ level compared to the $4\text{P}_{1/2}$ level, due to which the atoms will experience a lower effective trapping potential from the lattice during the cooling process, especially since excited states above 4P experience very weak trapping potentials at the lattice parameters used in our experiment.

Our results are particularly relevant in light of single-site high-resolution imaging schemes, where laser cooling is employed to generate the photons required for imaging. Raman sideband cooling has been shown to be particularly useful for this purpose as it not only produces the necessary fluorescence but also keeps a large atomic fraction in the ground state of the lattice sites, facilitating the production of low-entropy many-body quantum states [16, 50]. Moreover, the spatial resolution $r$ of an objective scales as $r = \frac{\lambda}{2NA}$, where $\lambda$ is the wavelength of light used for imaging and $NA$ is the numerical aperture of the objective. In the case of $^{39}$K, the wavelength corresponding to the 4S→5P transition is 1.9 times smaller than the 4S→4P transition wavelength, which implies a 1.9 times better resolution for the shorter wavelength for the same $NA$.

As a first step towards fluorescence imaging with 404-nm light, we directly collect the blue photons emitted via the spontaneous decay from the $5\text{P}_{1/2}$ level. An atom in the $5\text{P}_{1/2} |F = 1, m_F = -1\rangle$ level will decay by emitting a blue photon with 15% probability, while the remaining decay happens via the radiative cascade shown in Fig. 1(a). As an independent measurement, we repeat our experimental sequence, but with the polarizer beam at 0.9 MHz detuning, intensity of 2.9 mW/cm$^2$ and an exposure time of 20.5 ms (longer exposure time is not possible in our experiments since off-resonant scattering of light due to the closely detuned lattice light limits the lifetime). The detuning and exposure time were chosen to maximize the fluorescence signal. At the optimum parameters for cooling, most of the atoms end up in the dark state quickly and do not produce a strong fluorescence signal. The fluorescence was collected by an objective with a $NA$ of 0.16 and sent through a short-pass filter to reject the 767-nm light from the lattice, repumper and the radiative cascade before being sent to a CCD chip. Our measurement is shown in Fig. 6. The imaging optics, being designed for 767

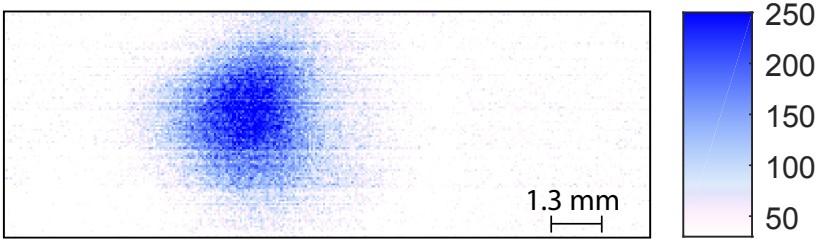

Figure 6: Blue photons collected from $1 \times 10^8$ atoms during an exposure time of 20.5 ms. The image is an average of 10 individual images. Each image is the difference of an image taken when the polarizer is on and a background image with the polarizer off. Pixel strength denotes fluorescence intensity. The polarizer intensity is 2.9 mW/cm$^2$ corresponding to 115 μW of power.

and 852 nm, are not anti-reflection coated for 404 nm, making the fluorescence collection process very lossy (major losses come from two mirrors used to deflect the light onto the CCD chip). Furthermore, our camera has a quantum efficiency of only 30% at 404 nm. By making a rough estimate of our losses, we determine from our measurement a lower bound of the scattering rate as 2.2 kHz per atom. This would correspond to a detection rate of 440 photons/atom per second for an $NA$ of 0.8, which is also a lower bound since it is calculated from the lower bound of the scattering rate. At this detection rate, to reach the photon counts of 750 to 1000 photons/atom required for high fidelity detection in existing high-resolution imaging techniques [16, 18], an exposure time of about 2.5 s will be sufficient. At $\delta_P = 4.9$ MHz, where cooling is only marginally less efficient than the optimum detuning of $\delta_P = 6.9$ MHz, an objective with a $NA$ of 0.8 would enable a theoretical detection rate of 192 photons/atom per second for a polarizer intensity of 2.9 mW/cm$^2$. At this detection rate, an exposure time of about 5 s will be required to obtain about 1000 photons/atom. In a far-detuned, deep optical lattice with minimal heating due to off-resonant excitations, it may be possible to further adapt our scheme to operate in a regime even better suitable for imaging. For a typical 1064-nm lattice, the anti-trapping nature of the 4P excited states of $^{39}$K and the strong light shifts associated with the large intensities employed in far detuned lattices would have to be taken into account [16]. An alternative would be a 'magic wavelength' lattice, which provides the same trapping potential to a ground-excited state pair. In addition, Raman couplings in existing non-degenerate Raman sideband cooling schemes for high-resolution imaging are provided by additional Raman beams along the three axes.

## 4 Conclusions

Although it was not a priori clear that sub-Doppler cooling could work via an $nS \rightarrow (n+1)P$ transition in alkali atoms, we have successfully demonstrated it via the 4S→5P transition of $^{39}$K at 404.8 nm. Using degenerate Raman sideband cooling, we polarize up to $5.6 \times 10^7$ atoms cooled to a temperature of 4 μK, a peak density of $3.9 \times 10^9$ atoms/cm$^3$, a phase-space density greater than $10^{-5}$ and an average vibrational level of $\langle v \rangle = 0.6$ in the lattice. We also demonstrate the feasibility of fluorescence imaging using 404-nm light by directly measuring the scattering rate of blue photons. The observation of cooling and significant scattering of shorter wavelength light on the $nS \rightarrow (n+1)P$ transition opens up the possibility of utlizing such transitions for single-site high-resolution imaging with significantly better spatial resolution. Since the level structure and cascade routes for K are similar to those in Na, Rb, Cs and Fr [39],

sub-Doppler cooling schemes on the $nS \rightarrow (n+1)P$ transitions are likely to work for all these elements.

## 5    Acknowledgements

We thank E. Kirilov, H. Ritsch, G. Anich and B. Santra for fruitful discussions and P. Weinmann for assembling the blue laser setup.

**Funding information**    We acknowledge support by the Austrian Science Fund (FWF) within the DK-ALM (W1259-N27). We gratefully acknowledge funding by the European Research Council (ERC) under Project No. 278417 and by the FWF under Project No. I1789-N20 (joint Austrian-French FWF-ANR project), under Project No. P29602-N36, and under Project No. Z336-N36 (Wittgenstein prize grant).

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
