# Peer review of "Sub-Doppler laser cooling of $^{39}$K via the 4S$\to$5P transition"

_SciPost Physics, doi:SciPost Phys. 6, 047 (2019)_

## Round 1 · Referee Report · Anonymous (Referee 1) · 2018-11-21

Strengths

1-: Experimental results on Raman-side band cooling using an low wavelength transition, which was never tested before.
2-: Potential application to single atom imaging with high resolution.

Weaknesses

1-: Interest restricted to the specific field of atom optical cooling.
2-: Some detailed points could be improved (see report)

Report

The paper deals with the the realization of Raman side-band cooling of 39K atoms using the 4S->5P transition at 405 nm. Previous studies have been restricted to more standard schemes using the D1 transition at 770 nm. The difficulty with optical pumping at 405 nm is linked to a more complicated cascade to the ground states manifold inducing a larger depolarization of the atoms during the cooling cycle. Nevertheless, the cooling scheme is proven to work. The achieved temperature is 4 microK, which is comparatively higher than the temperature using the D1 transition. The important result here is the fact that light at 405 nm, with a measured decent scattering rate can allow for better single-site-imaging, although this is not done in this work. Generally speaking the paper is well written. The discussion is rather technical with the involvement of several atomic levels. This level of details is however useful to understand the difference with Raman side-band cooling at 770nm. The paper is more addressed to a community of specialists of laser cooling and trapping. The results are however new and interesting for this community. Although I am not really aware of the publication policy of SciPost Physics, I would recommend this article for publication after the following comments/changes are taken into account.

Requested changes

1-Several graphs are presented as a function of cloud size after time of flight. This makes the reading difficult and unclear. It would more direct to have an estimation of the temperature (which is only given in the text for the lowest temperature). I understand that there can be issues with the initial size or distorted expansion.

2-In the comment of figure 5a, it is written ‘We observe optimal cooling performance at 6.9 MHz blue detuning’ and ‘we observe a similar atom number as for a commensurate blue detuning but significantly higher temperature’. I must say that looking at the figure, I do not really find back the value of 6.9 MHz neither do I see a pronounced asymmetry between blue and red detuning at low detuning. Please clarify.

3-The main argument for the importance of the paper is linked to in-situ high resolution imaging. As far as I understand, here long exposure is impossible due to a small lattice detuning ? Is changing to a far-detuned optical lattice the only necessary change ? Could you give typical numbers for a 1064 laser as an example ? Are problems linked to the lattice depth or to the Raman coupling to be expected ? More details would be helpful to really motivate the prospective of high resolution imaging. What is the gain as compared to the microscopes already working with potassium 40 and EIT cooling ? A release of the contrains on the numerical aperture ?

4-Figure 6 does not add meaningful information. It can be removed.

5-In the conclusion, could you comment on the possibility to use similar methods with other atomic species (isotopes of potassium, alkali, more exotic atoms) ?

  • validity: top
  • significance: good
  • originality: good
  • clarity: high
  • formatting: good
  • grammar: excellent

Author:  Govind Unnikrishnan  on 2019-02-21  [id 443]

(in reply to Report 2 on 2018-11-21)
Category:
answer to question

We thank the referee for his/her valuable comments. Please see our responses to the requested changes below.
1-For the sake of clarity, as pointed out by the referee, we have changed Fig. 4 (a) to plot the temperature vs the cooling time in place of cloud size vs cooling time. This figure shows a direct measurement of the temperature of the cloud when all other parameters are tuned to their optimum values indicated in Fig. 3. Furthermore, we now mention in the manuscript that a temperature of 4 uK corresponds to cloud radius of about 1.4 mm after a fixed TOF expansion to aid the interpretation of the other figures.
2-We have added a vertical bar in the figure to indicate the value of 6.9 MHz for more clarity. At small detunings, cooling is strongly compromised due to the strong broadening and large light shift of the energy levels by near resonant light, for both red and blue detuned light (as reflected in the reduced atom number). For detunings more than +-5 MHz where there is reasonable atom number, we see a clear asymmetery between the red and blue detunings. We have now included this point in the manuscript.
3-As pointed out by the referee, long exposure time is not possible in near detuned lattices as used in our case. We now mention this explicitly in the manuscript, as this was also pointed out in Report 4, point 13.
On the other hand, changing to a far-detuned lattice will have different challenges. For example, in a 1064-nm lattice, the 4P excited states of K experience strong anti-trapping potentials. A large scattering rate from the optical pumping beams can hence lead to increased heating and diffusion in such lattices [16]. This issue has been tackled by keeping the optical pumping beams far away from resonance as in Ref. [16]. Another factor that needs to be considered would be the strong light shifts, which would arise when using the high intensities required to get good trapping conditions in far-detuned lattices. The differential light shift to the levels involved would need to be taken into account. The use of a ‘magic wavelength’ lattice, which provides the same trapping potential to the ground and excited state, may prove useful in this case. Furthermore, many existing non-degenerate Raman sideband cooling schemes require additional Raman beams to induce Raman transitions, requiring a suitable change of frequencies of both the polarizer and repumper light fields. We now discuss these points in the last paragraph of Section 3.2.
The main advantage of this scheme as compared to existing microscopes would be a significant release on the constraints on the numerical aperture provided by the shorter wavelength.
4-Figure 6 shows the 404.8 nm fluorescence collected from the atoms. We use this figure to experimentally determine the scattering rate. We think that from an experimental perspective, it is an important step towards high-resolution imaging with shorter wavelength transitions.
5-Since the level structure and cascade routes for potassium is similar to those in Na, Rb, Cs and Fr [39], sub-Doppler cooling schemes on the nS->(n+1)P transitions are likely to be possible for all these elements. We have added this comment to our conclusion.

---

## Round 1 · Referee Report · Anonymous (Referee 2) · 2018-12-3

Strengths

  1. Demonstration of degenerate Raman side-band cooling using shorter wavelength polarising beam which opens up the possibility of higher resolution in-situ optical lattice imaging.
  2. Well written and explained.

Weaknesses

  1. Lacking some details that would be helpful in assessing feasibility of high-resolution imaging (see below).

Report

The manuscript describes the demonstration of degenerate side-band cooling of 39K using the 405nm 4S to 5P transition to (re)polarise the atoms. The overall cooling achieved is not as good as when using the D1 line [33] but the main advantage (selling point) of this approach is that the shorter wavelength would lead to greater optical resolution if this technique were applied to single-site in-situ fluorescence imaging of an optical lattice. The manuscript is generally well written and explained with an appropriate level of detail and I only have a few short comments/questions (listed in the 'Requested changes' section but not all necessarily requiring changes).

Requested changes

  1. This is maybe rather pedantic but technically doesn't the cooling coming via the 4S1/2 to 4P3/2 transition as this is where the majority of the Raman coupling will come from? Rather, it is just the polariser beam which is using the 4S1/2 to 5P1/2 transition which repolarises the sample but doesn't cool it?

  2. It would have been helpful to have the values of the magnetic bias field used.

  3. How good is the assumption of thermal equilibrium when converting T to <\nu>?

  4. [My most substantive point] (a) It would have been nice to see some data (or at least some discussion) on how the optimum detuning of the polarizer beam varies with its intensity. This seems particularly relevant when estimating the exposure times needed for imaging on the bottom of page 12/top of page 13. If a substantially higher intensity (than 2.9mW/cm^2) could be used, would it be possible to get a much improved scattering rate without a loss of cooling performance? (b) Related to this, I couldn't make sense of the change in scattering rate from 440 photons/atom/s to 192 photon/atom/s on going from a detuning of 0.9 to 4.9MHz, given the linewidth of the transition a naive calculation would give a much large reduction?? (c) It would be helpful to give the intensity of the polarizer beam as well as its power in Fig 5.

  • validity: top
  • significance: good
  • originality: good
  • clarity: high
  • formatting: excellent
  • grammar: excellent

Author:  Govind Unnikrishnan  on 2019-02-21  [id 444]

(in reply to Report 3 on 2018-12-03)
Category:
answer to question

We thank the referee for his/her valuable comments. Please find our responses below. 1. The Raman transitions do not lead to cooling by themselves. The Raman transitions are between degenerate energy levels. Cooling happens whenever the atoms travel down the non-degenerate vibrational levels in the F=2, mF=-1,-2 manifolds, which occurs only in combination with the polarizer. In principle, this can happen via the direct decay from the 5P1/2 level to the ground state or after a radiative cascade to the 4P levels from where a decay occurs to the ground state. When directly using the D1 line for the polarizer, only the F=1, mF=-1 state in the 4P1/2 is populated since we use a σ- polarized light field for the polarizer. This would mean an eventual dominant decay into the F=2, mF=-2 dark state of the ground state. However, the radiative cascade populates multiple Zeeman states of the F=1 hyperfine excited state, leading to a less dominant decay into the dark state. Hence, it is not equivalent to the cooling mechanism which occurs when using a polarizer on the D1 line. It is because of these reasons that we have used the terminology “cooling via the 4S-5P” transition.

  1. We use bias fields values very close to those in Ref. [33]. We have mentioned this now in the first paragraph in Section 3.2.

  2. The densities reached after our cooling procedure is of the order of a few $10^9 cm^{-3}$, which is still very dilute (for comparison, our BEC densities are of the order of $10^{13} cm^{-3}$). Hence, we do not expect multiple atoms to occupy the same lattice site. The question of thermal equilibrium between atoms in the same lattice site does not arise as a consequence. However, we do expect atoms to occupy vibrational levels in separate lattice sites commensurate with their kinetic energies. This assumption has previously been used to calculate $<\nu>$ for laser cooled ions, where the average <> can be over time or over an ensemble [21,48]. In our case, each experimental run involves the realization of an ensemble of atoms trapped in multiple lattice sites.

  3. (a) First, we clarify that we have done two independent measurements: 1) Cooling: The polarizer intensity and detuning of 2.9 mW/cm$^2$ and 6.9 MHz, respectively, was the combination we obtained for the best cooling performance – in terms of temperature and atom number. The duration of our sequence was a 6.35 ms. 2) Fluorescence collection: As an independent measurement, we show that it is possible to collect blue fluorescence by carrying out the same experimental sequence but with the polarizer beam at 0.9 MHz detuning and an exposure time of 20.5 ms. The detuning and exposure time were chosen to maximize the fluorescence signal. At the optimum parameters for cooling, most of the atoms end up in the dark state quickly and do not produce a strong fluorescence signal. Since the measurements are independent, the optimum detuning and intensity used for cooling are not relevant for estimating the exposure times mentioned in the last paragraph of Section 3.2.
    At the optimal detuning for cooling (6.9 MHz), we found that the cooling performance increased with increasing polarizer intensity. We work at the maximum intensity we can achieve (2.9 mW/cm$^2$). It may be possible to achieve an improved cooling performance with an even higher polarizer intensity. However, limiting factors could be the radiation pressure due to the optical pumping, which can push the atoms out of the lattice [23]. A higher repumping rate may be required, which can affect the final temperatures achieved. Moreover, for a 1064-nm lattice, the 4P excited states are strongly anti-trapping for potassium. A large scattering rate of the optical pumping beams can hence lead to increased heating and diffusion in such lattices [16].

We clarified that the fluorescence measurement is independent and included our justifications in the last paragraph of Section 3.2.

  1. (b) The difference between the two scattering rates is not as large as one would expect from a theoretical calculation since the value of 440 photons/atom/s is obtained from our measured scattering rate of 2.2 kHz, which is only a lower bound as mentioned in the manuscript (we believe that we underestimate our overall losses). The value of 192 photons/atom/s, on the other hand, is the detection rate calculated from the theoretical scattering rate and a NA of 0.8 at 4.9 MHz detuning. In the manuscript, we have now emphasized that the value of 440 photons/atoms/second is a lower bound.

  2. (c) We have now provided both the intensity and power of the polarizer beam in the caption of Fig. 5.

---

## Round 1 · Referee Report · Anonymous (Referee 3) · 2018-12-14

Strengths

The authors present an experimental study demonstrating sub-Doppler cooling of 39K in an optical lattice. In contrast to previous approaches, this is accomplished through degenerate Raman side-band coupling in a closely-detuned optical lattice, combined with an optical pumping scheme that results in the atoms being shelved in a dark state once they are cooled to the vibrational ground state of the lattice. Although the method, by virtue of using blue-detuned light as opposed to red-detuned light doesn’t reach as low of final temperatures, a notable reduction of the temperature and corresponding increase in phase-space density is accomplished over a few milliseconds.

The technique appears novel and is technically quite interesting, especially in regards to the applications for single-site resolved imaging, where tight resolution requirements imposed by optical lattice spacings would benefit from using shorter imaging wavelengths.

Weaknesses

The paper is suitable for publication, but I found a number of issues with the presentation and explanation that made the text unnecessarily complex and somewhat difficult to follow. I strongly recommend that the authors make the following adjustments to the presentation to improve the readability of the manuscript.

A general comment is that without prior familiarity with the Raman side-band cooling scheme, it is not clear that the Raman coupling is intrinsic to the lattice, in the presence of the magnetic field. After reading the references I understand this now, but the text did not walk the reader through the technique with sufficient detail to understand the process.

Report

1. The first sentence of the abstract should refer to the optical lattice and applied magnetic field, along with the optical pumping beam, since these are key to the process.

2. The lattice can be introduced in the 6th sentence of the 1st paragraph of section 2.1, where the Raman coupling is mentioned, to explicitly mention that the coupling is due to the lattice.

3. In sentence 3 of of the 1st paragraph of section 2.1, where the lattice is first mentioned, what is the lattice wavelength? Although this is stated later, it would make sense to have this information here. Furthermore, what is the configuration of the beams? Is it the same as Ref 22? Please specify this information.

4. The sentence “A \sigma_{-} polarised beam tuned tp the 4S—>4P…” I think this sentence would be better suited as a footnote (to avoid confusion with the polariser beam).

5. It would be clearer to start the discussion of the cooling process by specifying the initial state — presumably |F=2, mF = -2, v>, ending up in |F=2, mF = -2, v-2>, which would make the process more apparent.

6. The authors should consider switching the order of sections 2.1 and 2.2. There is considerable information in section 2.2 that is relevant for understanding the cooling scheme and Fig. 1 — it would be more sensible for this information to precede section 2.1 Figures 1 and 2 should also switch order as a consequence.

7. Figure 1: It seems like the broadening of the transitions is somewhat of a minor detail (mostly relevant to the later discussion of the cooling results). It doesn’t seem like it should be indicated on the figure, as it gives an impression that the broadening is actually important to the cooling scheme (rather than detrimental).

8. It would be useful in the caption of Fig. 1 to mention again that the Raman coupling between states is provided by the lattice beams.

9. Section 3.1: I’m not sure why the power and intensity of the lattice and repumper are the relevant parameters, it would be useful to specify the wavelengths again here to avoid confusion. Perhaps it would be better to specify the lattice depth here? I don’t think the repumper intensity is all that important, assuming it is not saturating?

10. Section 3.2: How long is the cooling applied for? The timing of each of the other steps is mentioned, it would be clearer to mention the typical cooling period in this location in the text.

11. Section 3.2: I also noted that the measurement procedure seems to imply that there are extra components other than |F=2, mF = -2> that are imaged — but not cooled? Can this be addressed in the text, especially if residual components might limit the final temperatures? What fraction of atoms end up in the desired state?

12. Fig.4(b) this panel seems unrelated to panel (a), so it should be a separate figure. I’m not sure of its value anyways, as this is presumably just a standard measurement done when calculating the lattice frequency. It is fine to just state the result in the text only.

13. In describing the fluorescence measurements, I noted the relatively short exposure times, compared to what would be needed in the single-site resolving experiments. What limits the exposure time here? Is there some issue with lifetime in the lattice due to the closely-detuned lattice wavelengths? What is the lifetime? This should be addressed, as I wondered if there is some technical issue that is not being identified (and would be important for the multi-second exposures discussed in the text).

Requested changes

In addition to the above changes there were some minor issues:

Section 1. Paragraph 2, “Doppler temperatures in optical lattices”

Section 2.2, Paragraph 4, “levels are 286 nm, 454 nm, ….”

Section 3.2, Paragraph 1, “detunings of -9 MHz and 6.9 MHz” (unit was left off)

Section 3.2, Paragraph 2 “We estimate a systematic..”

I would recommend a further read through the text for remaining typos and any outstanding grammar issues.

  • validity: high
  • significance: good
  • originality: good
  • clarity: ok
  • formatting: reasonable
  • grammar: excellent

Author:  Govind Unnikrishnan  on 2019-02-21  [id 446]

(in reply to Report 4 on 2018-12-14)
Category:
answer to question

We would like to thank the referee for his/her comments. Please find our responses below. 1. As suggested, we have included these points in the abstract.

  1. We have changed the 6th sentence of the first paragraph of section 2.1 as follows to explicitly indicate that the Raman couplings are due to the lattice: “The frequency and polarizations of the lattice light beams are chosen such that there are significant Raman couplings due to the lattice light between the states that are brought into degeneracy by the magnetic field.”

  2. In section 2.1, the idea is to describe degenerate Raman sideband cooling in a more general context using the example of 39K. We have made changes so that this is more apparent. In this section, we mention that the lattice light frequency is chosen to provide good Raman couplings and provide information that are specific to our experimental sequence, such as the lattice configuration and detuning, in Section 3.1.
    We use the same beam configuration as in Ref. [33], as mentioned in Section 3.1.

  3. We have changed our sentence as follows to avoid confusion: “A $second$ $ \sigma^- $-polarized beam (so called ``repumper), which is tuned to the 4S$ _{1/2}$$ F=1 \rightarrow$4P$ _{1/2} $ $ {F=2} $ transition , depletes population in the $F=2 $ manifold"

  4. As suggested, we now include the starting state |F=2, mF = -2, v> in our discussion in Section 2.1.

  5. We agree with the referee. The order has been switched.

  6. The broadening is not always detrimental. At the intensities used in our experiment, it makes the cooling scheme more robust as it allows appropriate Raman transitions to take place in a broader range of magnetic field values than what would be possible without broadening.

  7. We have included this information in the caption.

  8. We report all experimental parameters in section 3.1 for completeness. We have now included the lattice depth as well in this section. Lamb-Dicke parameters are also provided in section 3.2. The repumper intensity should be large enough to deplete the population in the F=1 manifold, which would otherwise result in heating. We include the powers and intensities as these are experimentally relevant parameters

  9. Cooling takes place during the entire cycle, but largely during the ramping of the offset magnetic field to sweep through different vibrational levels.

  10. The other spin components are imaged after Stern-Gerlach separation, as was also done in Ref. [33] to determine the fraction of atoms in the desired state. In our case, our total atom number was not enough to have a significant fraction in the other spin components. On the other hand, we do not think that the other spin components will limit the final temperatures since our samples are still quite dilute to allow for any significant interaction to take place.

  11. Fig. 4 (b) is used to extract the lattice frequency required to obtain the mean vibrational quantum number plotted in Fig. 4 (a). As opposed to typical measurements [32], we observe a strong broadening of the modulation spectrum due to strong Raman couplings in our system. We give Fig 4 (b) as a separate figure to highlight these points.

  12. In our experiments, off-resonant scattering of light due to the closely detuned lattice light limits the lifetime (the off-resonant scattering rate of the lattice being close to 4.6 kHz). We now mention this explicitly in the manuscript.

Responses to requested changes: 1. Section 1. Paragraph 2, “Doppler temperatures in optical lattices”

 -We have made this correction.

  1. Section 2.2, Paragraph 4, “levels are 286 nm, 454 nm, ….”

 -Since we refer to $the$ ionization threshold, which is singular, we now write the sentence as: “The ionization threshold from the levels 4S$ _{1/2} $, 4P$ _{1/2} $, 4P$ _{3/2} $, 5S$_{1/2} $, 3D$ _{3/2} $, and 5P$ _{1/2} $ $is$ 286 nm, 454 nm, 455 nm, 715 nm, 742 nm and 972 nm away, respectively."

  2. Section 3.2, Paragraph 1, “detunings of -9 MHz and 6.9 MHz” (unit was left off)

 -We have now included the units.

  3. Section 3.2, Paragraph 2 “We estimate a systematic..”

 -We rewrite the sentence as “We estimate systematic errors less than 25% in our temperature and atom number measurements.”

---

## Round 2 · Referee Report · Anonymous (Referee 3) · 2019-2-25

Report

I am satisfied that the authors have satisfactorily addressed my comments from the initial round of review, and recommend publication.

---

## Round 2 · Referee Report · Anonymous (Referee 2) · 2019-2-26

Report

I am generally happy with the authors response to my questions. I only have a few minor suggestions.

(a) In section 2.2, the sentence beginning 'A second \sigma^- polarised beam (so called "repumper)...' is confusing as this beam doesn't deplete the F=2 manifold but rather keeps everything in the F=2. Also it appears in the middle of the paragraph and just gets in the way of the flow of the argument. I would move it to a footnote (as suggested by one of the other referees) or at least to the end of the paragraph where you could say 'Note that an sigma polarised repumper beam tuned to the ... transition (see fig 3) is also required to prevent population of the F=1 groundstate.'

(b) Both the power and intensity of the polariser beam are now given in Fig 6 but not in Fig 5 - it would be good to have in both places to make it clear that the same power was used in both cases.

(c) On p13, it would be helpful to replace the sentence beginning 'At \delta_p = 4.9MHz, where cooling is only marginally less efficient,...' with 'At \delta_p = 4.9MHz, where cooling is only marginally less efficient than the optimum detuning of \delta_p = 6.9MHz ,...'. The previous thing that was referred to was the \delta_p = 0.9MHz case which as I understand it is not what is being compared to.
  • validity: -
  • significance: -
  • originality: -
  • clarity: -
  • formatting: -
  • grammar: -

Author:  Govind Unnikrishnan  on 2019-03-24  [id 472]

(in reply to Report 2 on 2019-02-26)
Category:
answer to question

We thank the referee for the additional comments. We have made all the three recommended changes.

(a) In section 2.2, we moved the relevant sentence to the end of the paragraph. We now say, "Note that a second $ \sigma^- $-polarized beam (so called ``repumper"), which is tuned to the 4S{1/2} F=1 ->4P1/2 F=2 transition (see Fig. 3), depletes population in the 4S 1/2 F=1 manifold", at the end of the paragraph.

(b) Both the power and intensity of the polarizer beam is now mentioned in all the three relevant figures (Fig. 4 (a) ,5, and 6).

(c) As suggested, we have now made it explicit that the comparison is made to the detuning at 6.9 MHz.

---

## Round 2 · Referee Report · Anonymous (Referee 1) · 2019-3-12

Report

I am now fine with publication.

---

## Round 2 · List of Changes

The changes are indicated in the responses to the referee reports.

---

## Round 3 · Author Response

We thank the referees for their comments. We have made all three changes recommended in Report 2.

---

## Round 3 · List of Changes

Please see reply to Response 2.

---

## Editorial Decision

published